# Understanding the Impact of Client Heterogeneity on Ordinal Classification in Federated Medical Image Analysis

**Valentina Corbetta**[1,2,3] ⦿            V.CORBETTA@NKI.NL
**Regina Beets-Tan**[1,3] ⦿            R.BEETSTAN@NKI.NL
**Jaime S. Cardoso**[4,5] ⦿            JSC@FE.UP.PT
**Wilson Silva**[1,2] ⦿            W.J.DOSSANTOSSILVA@UU.NL

[1] *Department of Radiology, The Netherlands Cancer Institute*

[2] *AI Technology for Life, Utrecht University*

[3] *GROW Research Institute for Oncology and Reproduction, Maastricht University*

[4] *Faculdade de Engenharia, Universidade do Porto*

[5] *INESC TEC*

**Editors:** Accepted for publication at MIDL 2025

## Abstract

Deep learning methods have shown remarkable success in medical image classification, aiding in early disease detection and treatment. Many of these tasks, such as cancer staging or risk stratification, exhibit an inherent ordinal structure; however, existing solutions often reduce them to binary or purely nominal classifications, ignoring the valuable ordering information. Simultaneously, privacy and regulatory concerns have spurred the adoption of Federated Learning (FL), enabling collaborative model training without centralising sensitive patient data. Yet, FL in real-world medical scenarios faces significant challenges arising from heterogeneous client data, particularly when institutions differ widely in case severity or label distribution. In this work, we conduct the first in-depth study of Federated Ordinal Learning (FOL), introducing ordinal classification paradigms into FL pipelines and systematically evaluating their performance under increasing levels of data heterogeneity. We assess the benefits of ordinal classification within four FL frameworks: standard Federated Averaging (FedAvg) and three heterogeneity-focused approaches (FedProx, MOON, and FedALA). Our experiments reveal that ordinal methods can effectively maintain class ordering information even when institutional data exhibit severe imbalance or missing classes, offering valuable insights for developing robust, privacy-preserving AI systems in medical imaging. However, ordinal approaches still suffer from performance degradation in highly heterogeneous FL settings, underscoring the need for dedicated research on FL methods that explicitly account for ordinality.

**Keywords:** federated learning, data heterogeneity, ordinal classification

## 1. Introduction

In recent years, Deep Learning (DL) models have significantly advanced automatic medical image classification, particularly aiding in early detection and treatment of diseases like cancer, which is crucial for reducing mortality rates (Cai et al., 2020; Duffy et al., 2021; Murtaza et al., 2020). Many of these classification tasks are multi-class problems with an inherent ordinal structure, where classes follow a natural order of severity. However, much of

Code is available at: https://github.com/Trustworthy-AI-UU-NKI/Federated_Ordinal_Learning

the existing literature on medical image classification often approaches these tasks as binary classifications or as multi-class nominal classifications (Lei et al., 2020; Arvaniti et al., 2018). Binary approaches tend to overlook indeterminate or ambiguous classes, essentially ignoring cases that fall between the extremes (e.g. dubious cases between benign and malignant, healthy and pathological) and that are deemed uncertain by clinicians. Nominal methods, instead, fail to consider the natural order between classes, making results less reflective of label progression and leading to a loss of information. Ordinal classification blurs the lines between classification and regression tasks. Unlike nominal classification, where labels are distinct and unrelated, ordinal classification involves labels that possess an inherent order, similar to regression scenarios (Frank and Hall, 2001). Although the emphasis on ordinal classification within medical imaging remains relatively limited, there is noteworthy work in this area. Albuquerque et al. (2021) proposed an ordinal loss that promotes the probabilities of output to follow a unimodal distribution for the classification of cancer risk. In (Lei et al., 2022), a novel meta-ordinal regression forest method is proposed for medical image classification with ordinal labels, which combines neural networks (NNs) with a differential forest to capture the ordinal relationship. Le Vuong et al. (2021) combined an ordinal loss with a nominal classification loss to improve cancer classification.

As privacy concerns grow, the field of medical image analysis is increasingly adopting a Federated Learning (FL) paradigm (Van Panhuis et al., 2014; Rieke et al., 2020). FL allows multiple institutions to collaboratively train models without sharing sensitive patient data, making it an attractive approach in healthcare (Zhang et al., 2021). However, integrating FL with DL tasks presents challenges, especially due to data heterogeneity between institutions. For example, some hospitals may predominantly handle less severe cases, while specialised centres see more critical ones. This imbalance can hinder the global model's performance, as it may not receive a balanced representation of all classes. Various FL methods have been proposed to tackle the heterogeneity of clients from different angles. Parameter regularisation approaches (e.g., FedProx, (Li et al., 2020)) modify the local objective to include a proximal term, preventing local models from drifting too far from the global parameters and mitigating instability arising from skewed distributions. Representation-based solutions (e.g., MOON, (Li et al., 2021)) introduce contrastive objectives to align local and global feature spaces, reducing overfitting to domain-specific biases. Meanwhile, adaptive local training strategies (e.g. FedALA, (Zhang et al., 2023)) adjust aggregation weights or learning rates to accommodate varying client data distributions.

Although the issues of class imbalance and data heterogeneity at the client level have been investigated in general DL tasks, including nominal classification, it has not been studied in the context of ordinal classification, where the use of ad hoc ordinal paradigms is essential for accurately learning the ordered relationships between classes.

Our aim is to analyse the impact of client heterogeneity in Federated Ordinal Learning (FOL). We hypothesise that, despite the presence of missing or under-represented classes at the client level, ordinal losses can still effectively capture and learn ordinal relationships.

Our contributions in this analysis are as follows: (1) Novelty of ordinal FL: to the best of our knowledge, this is the first study to introduce ordinal classification in federated learning settings for medical image analysis. (2) Systematic heterogeneity assessment: we rigorously assess the performance of FOL under varying, increasingly heterogeneous data partitions, providing insights into how ordinal methods behave across diverse institutional

settings. (3) Evaluation of state-of-the-art FL approaches with ordinality: we integrate two ordinal learning paradigms, one parametric and a non-parametric one, into four FL algorithms—standard Federated Averaging (FedAvg) (McMahan et al., 2017) and three heterogeneity-oriented frameworks (FedProx, MOON, and FedALA)—to highlight their effectiveness in non-IID, real-world conditions. (4) We perform extensive experiments on the CSAW-M dataset (a benchmark for ordinal classification) (Sorkhei et al., 2021), simulating different degrees of heterogeneity via a Bernoulli-Dirichlet sampling strategy.

## 2. Materials and Methods

### 2.1. Ordinal Classification

In ordinal learning literature, a vast amount of research focuses on encouraging a unimodal distribution in the posterior probability $q(\mathbf{y}|\mathbf{x})$, where $\mathbf{y}$ represents the target class labels and $\mathbf{x}$ represents the input features. This is typically achieved through two types of learning paradigms: parametric and non-parametric (Niu et al., 2016; Frank and Hall, 2001). Parametric approaches typically enforce unimodality on the posterior distributions by applying a single penalty across all labels. In contrast, non-parametric methods avoid constraining the learnt representation to a single parametric family, offering greater flexibility. We evaluate one approach from each category: the Binomial unimodal regularised cross entropy loss, from now on referred to as Binomial Cross-Entropy (BCE) (parametric) (Liu et al., 2020), and Ordinal Encoding (OE) (non-parametric) (Frank and Hall, 2001).

**Binomial Cross-Entropy (BCE)** In the standard one-hot setting, the label distribution for class $l$ is given by $q(i) = \delta_{i,l}$, where $i \in 0, ..., N-1$ (with N the total number of classes), and $\delta_{i,l}$ is a Dirac delta that equals 1 only if $i = l$ and 0 otherwise. Traditional label smoothing replaces this delta distribution in the Cross-Entropy (CE) loss with a convex combination of $\delta_{i,l}$ and a uniform distribution over all classes $i$ (Zou et al., 2019). Formally,

$$q'(i) = (1 - \eta)\delta_{i,l} + \eta\frac{1}{N} \tag{1}$$

where $\eta \in [0, 1]$ controls the smoothing intensity. As uniform smoothing does not explicitly account for the ordinal nature of the labels, BCE instead replaces the uniform distribution with a binomial distribution $p(i)$, unimodally centred on the ground-truth class $l$, thus acting as an ordinal-aware regularisation. The final target distribution becomes

$$q'(i) = (1 - \eta)\delta_{i,l} + \eta p(i), \tag{2}$$

$$\text{where } p(i) = \binom{N-1}{i}p^i(1-p)^{(N-1)-i}. \tag{3}$$

This softens the one-hot label while encouraging adjacent classes to receive higher probabilities than distant ones.

**Ordinal Encoding (OE)** Classes are encoded as a cumulative distribution. Let $l$ be the ground-truth class for the $n$th sample. We define

$$y_{n,i} = \begin{cases} 1, & \text{if } i < l, \\ 0, & \text{otherwise,} \end{cases} \quad i \in \{0, \ldots, N-2\}. \tag{4}$$

As a result, the model's output is expected to increase monotonically with $i$, reflecting the cumulative nature of ordinal labels. One key advantage is its independence from the particular training objective: the encoding alone promotes ordinality. At inference time, to obtain the standard probability $q_i$ of class $i$, we perform the following

$$q(i) = \begin{cases} 1 - \sigma(g_1), & \text{if } i = 0, \\ \sigma(g_{i-1}) - \sigma(g_i), & \text{if } 1 \leq i \leq N-2, \\ \sigma(g_{N-2}), & \text{if } i = N-1, \end{cases} \tag{5}$$

where $\sigma(x) = \frac{1}{1+e^{-x}}$ is the sigmoid function and $g_i$ are the logits in output of the classification model.

We use CE as our baseline method to evaluate the performance of the ordinal strategies. Although CE is the standard choice for multi-class classification, it only maximises the probability of the correct class, while ignoring relative probabilities among other classes, a limitation in ordinal problems where inter-class relationships carry important information.

## 2.2. Federated Learning Methodologies

We integrate BCE and OE into multiple FL methods, which are designed without ordinality in mind and thus trained with CE. We adopt FedAvg as our baseline FL algorithm, which computes a weighted average (by sample count) of locally trained parameters for each client. While FedAvg is straightforward and widely used, it can struggle when data are heterogeneous across clients. To address this limitation, we evaluate three heterogeneity-aware approaches:

1. FedProx, which introduces a proximal term in the local objectives to reduce client drift and stabilise global convergence;

2. MOON, which leverages contrastive learning to align local and global representations, thus mitigating the effects of non-i.i.d data;

3. FedALA, which adaptively aggregates the global and local models before each training round. Instead of fully replacing the local model with the global one, FedALA learns element-wise adaptive weights that selectively integrate only the most relevant information from the global model.

## 2.3. Dataset and experimental setup

In our study we use the CSAW-M dataset, comprising 10,020 mammography images, of which 9,523 are used for training and 497 for testing. The goal is to classify the degree of masking, i.e., how much a tumour is obscured by surrounding breast tissue, potentially hindering its detection via standard mammography. Masking levels range from 0 to 7. We selected this dataset because it was specifically designed to benchmark ordinal classification methods. Moreover, since all images originate from the same hospital, it allows us to

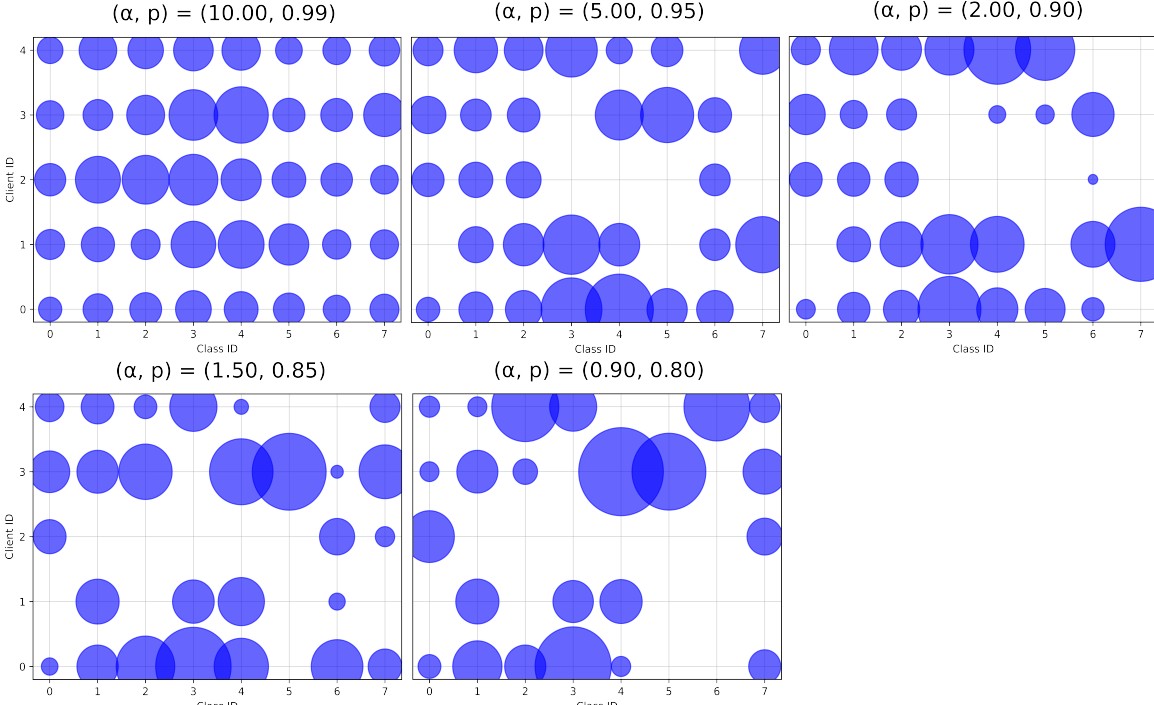

Figure 1: Bubble plots showing the distribution of class samples across five clients. Larger bubbles indicate more samples, with increasing data heterogeneity as $\alpha$ and $p$ decrease, generated for the train/validation split with seed $= 0$.

simulate a controlled level of heterogeneity in which the only source of variation is the label distribution among clients, rather than differences in patient population, acquisition protocol, or scanner type. To simulate data heterogeneity, we distribute the data unevenly across $K = 5$ clients, following the approach of Wu et al. (2023). A binary matrix $\Phi \in {0, 1}^{K \times N}$ is generated. Each element $\Phi_{k,i}$ is sampled from a Bernoulli distribution with probability $p$, indicating whether client $k$ has samples of class $i$ (1 for yes, 0 for no). For each class $c$, a Dirichlet distribution with parameter $\alpha$ is used to determine how the class samples are distributed among the clients that possess the class. We conduct five experiments, progressively lowering $\alpha$ and $p$ to increase the degree of heterogeneity. Bubbleplots showing the different sample distributions are depicted in Figure 1.

## 2.4. Ordinal Metrics

To assess model performance, we use three ordinal metrics: Mean Absolute Error (MAE), Uniform Ordinal Classification Index ($A_{uoc}$) Silva et al. (2018), and Kendall's $\tau_b$. For a detailed explanation of these metrics, we refer the reader to Appendix A. Additionally, we report Balanced Accuracy, a common metric in standard nominal classification, though we recognise it is not ideally suited for ordinal classification.

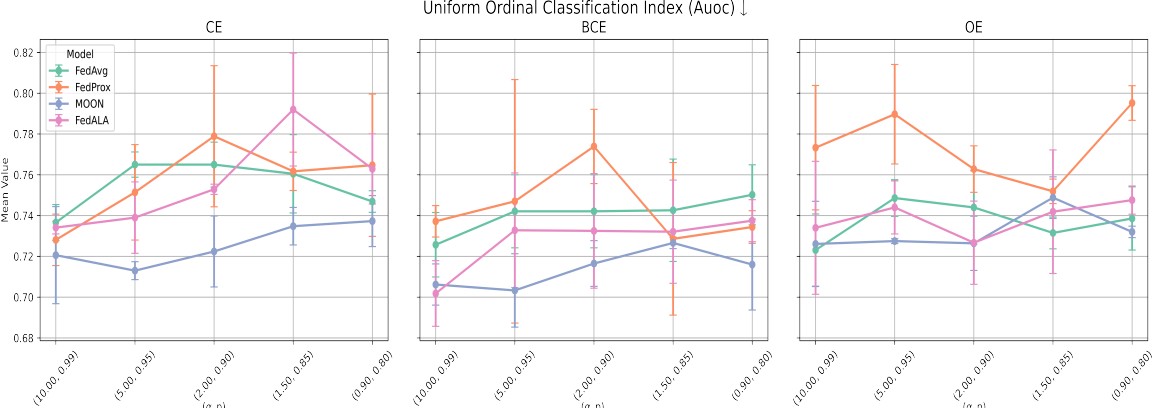

Figure 2: Performance trends in terms of Uniform Ordinal Classification Index ($A_{uoc}$) for the Federated Learning methods (FedAvg, FedProx, MOON, FedALA) with increasing data heterogeneity (x-axis: $\alpha$ and $p$ values), comparing: Cross-Entropy (CE), Binomial Cross-Entropy (BCE), and Ordinal Encoding (OE). Results are averaged and reported with the standard deviation across three runs.

## 2.5. Model Architecture and Training Settings

We train ResNet-34 (He et al., 2016) with ImageNet (Deng et al., 2009) pre-trained weights. Regarding pre-processing, we rely on the authors' pre-processed images, applying additional horizontal/vertical flips (each with probability 0.5), plus random rotations and colour jitter (each with probability 0.3). Models are trained with a batch size of 16. Regarding the hyperparameters, we fine-tune each FL method separately; detailed results of this tuning process, along with the final parameters, are presented in Appendix C. For each combination of learning paradigm, FL method, and heterogeneity configuration, we train three separate models, each using a unique train/validation split, and then average their performance on the global test set. The optimal model for each run was chosen by minimising $A_{uoc}$, the most complete of the ordinal metrics.

## 3. Results

Figure 2 illustrates the performance trends for the four FL methods and three learning approaches under increasing heterogeneity, evaluated using $A_{uoc}$. Overall, BCE and OE consistently outperform CE across all levels of heterogeneity. Interestingly, performance tends to decline most in intermediate heterogeneity. However, the drop in performance for the highest heterogeneity scenario might be mitigated by the fact that model hyperparameters are fine-tuned for this setting. Additionally, BCE and OE exhibit smaller performance gaps across heterogeneity levels compared to CE. OE demonstrates the lowest standard deviation across runs, showcasing the stability of its performance gap over CE.

When comparing FL methods, heterogeneity-aware approaches generally outperform FedAvg across all learning paradigms, except for FedProx. Notably, the effect of FedProx heavily depends on the hyperparameter $\mu$, which controls the contribution of the proximal

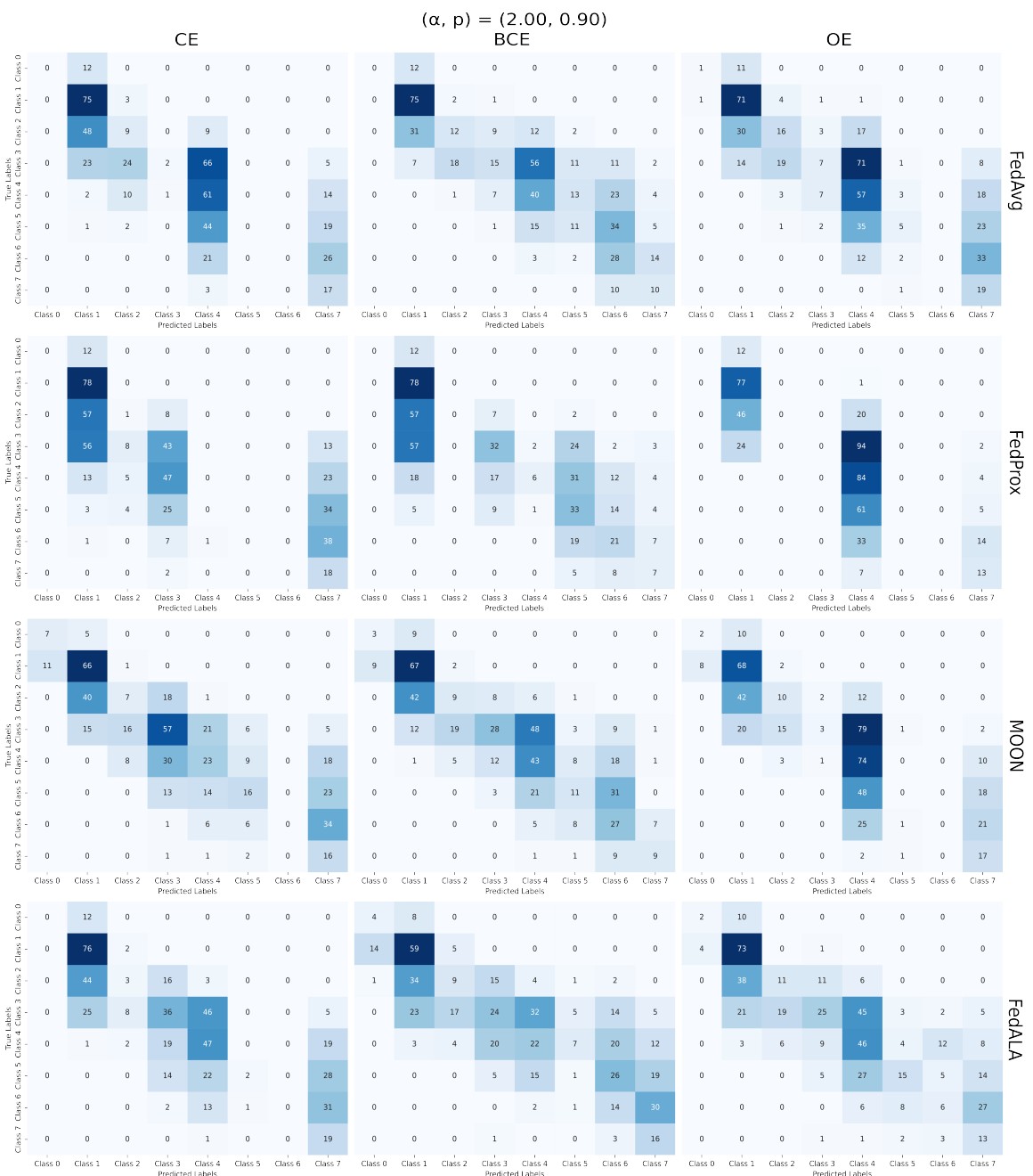

Figure 3: Confusion matrices for the $(\alpha, p) = (2.00, 0.90)$ train/validation split (seed = 0) for the Federated Learning methods (FedAvg, FedProx, MOON, FedALA) with Cross-Entropy (CE), Binomial Cross-Entropy (BCE), and Ordinal Encoding.

term in the local objective function. Since $\mu$ is fine-tuned exclusively for the highest heterogeneity setting due to computational constraints, it might not be optimal for the other heterogeneity levels. For FL methods integrated with the ordinal approaches, performance gains appear to stem primarily from the ordinal paradigms, as the performances of these FL strategies yield more similar results than CE, which is more sensitive to the choice of FL method. Equivalent plots for MAE, Kendall's $\tau_b$, and Balanced Accuracy are provided in Appendix D, Figure 4. MAE and Kendall's $\tau_b$ show trends consistent with those observed previously, although the differences are less pronounced for Kendall's $\tau_b$. In contrast, Balanced Accuracy does not clearly differentiate between CE and ordinal methods, as all three approaches yield similar performances. This indicates that ordinal methods maintain comparable overall classification accuracy while providing improvements specifically in terms of preserving ordinal relationships, as highlighted by the ordinality-informed metrics. The numerical results for all the metrics are reported in Tables in Appendix D.

To gain deeper insight into classification performance, we analyse the confusion matrices for the intermediate heterogeneity setting, which exhibits the worst performance (Figure 3).

Overall, we observe that CE struggles to maintain continuity along the diagonal, breaking ordinality. BCE consistently preserves diagonal continuity, ensuring better ordinal relationships between classes. Additionally, FedALA outperformes FedAvg on class 7, which is missing in four out of five centres.

Comparing the two ordinal approaches, OE is particularly affected by missing classes. Since OE learns the boundaries between classes, the heterogeneous scenario exacerbates its limitations: not only is class 7 absent in most centres, but either class 6 or class 5 is also missing, reducing the available information needed to define class boundaries. FedALA stands out as the only FL approach capable of mitigating this issue.

While both FedALA and MOON operate at the client level to align local models with the global one, FedALA employs learnable weights that dynamically scale the difference between the two models. This adaptability gives FedALA a distinct advantage over MOON's more static contrastive learning approach, where the closeness between global and local models is strictly controlled by the hyperparameters $\mu$ and $\tau$, which determine the contrastive loss contribution in the local objective function.

FedProx, on the other hand, performs poorly, likely due to its strong dependence on the hyperparameter $\mu$. When $\mu$ is not optimally fine-tuned, it hinders the learning process. It can be seen that for the highest heterogeneity scenario (Figure 5 in Appendix D) on which $\mu$ was fine-tuned, these effects are mitigated. Conversely, in the lowest heterogeneity setting (Figure 6 in Appendix D), where no classes are missing, ordinality remains intact across all methods, except for FedProx, which suffers from the aforementioned sensitivity to $\mu$. A more detailed analysis of hyperparameter tuning is provided in Appendix C.

For a direct comparison with centralized training, we provide baseline results trained on the lowest heterogeneity setting in Appendix D (Table 2), demonstrating that FedAvg achieves comparable or better performance. Moreover, a detailed assessment of computational runtimes (epoch and aggregation times) is reported in Appendix D, Tables 3 and 4, highlighting trade-offs between computational efficiency and methodological complexity.

## 4. Conclusions

This study is the first to explore Federated Ordinal Learning (FOL), integrating ordinal classification into FL for medical imaging. Our finding show that ordinal approaches (BCE, OE) improve robustness to missing classes and class imbalance, outperforming standard nominal classification (CE). However, despite their advantages, ordinal methods still suffer from performance degradation under high heterogeneity, highlighting the need for FL strategies that explicitly account for ordinality.

Our experiments highlight that current heterogeneity-aware FL strategies—such as Fed-Prox, MOON, and FedALA—require careful hyperparameter tuning. Notably, hyperparameters optimized for the highest heterogeneity scenario might not be optimal for lower heterogeneity levels. Future research should thus prioritize adaptive strategies that dynamically adjust hyperparameters based on varying heterogeneity, preserving ordinal relationships across diverse scenarios.

Moreover, our study leveraged the CSAW-M dataset, a carefully curated ordinal benchmark with consistent annotation protocols and minimal confounders. This choice facilitated a rigorous and controlled initial evaluation. However, extending our analysis to additional real-world datasets would further validate the generalizability of our findings. As part of future work, we plan to investigate other datasets, which would include different imaging modalities, diverse patient populations, and varied annotation practices, providing additional insights into FOL in practical clinical environments.

Lastly, while our experiments considered five clients, a number consistent with many recent federated learning studies in medical imaging literature, scaling to a larger federation could further strengthen the robustness of our conclusions.

Future work will therefore focus on: (1) exploring adaptive hyperparameter strategies that dynamically adjust based on heterogeneity, (2) integrating and adapting more advanced FL methods (e.g., FedSAM (Qu et al., 2022), FedSoup (Chen et al., 2023), FedRep (Collins et al., 2021), FedBABU (Oh et al., 2021)) to effectively leverage ordinal structures, and (3) expanding our evaluations to larger federations and multiple diverse datasets to robustly assess the scalability and generalizability of federated ordinal approaches.

## Acknowledgments

Research at the Netherlands Cancer Institute is supported by grants from the Dutch Cancer Society and the Dutch Ministry of Health, Welfare and Sport. The authors would like to acknowledge the Research High Performance Computing (RHPC) facility of the Netherlands Cancer Institute (NKI). This publication is part of the project "Ordinality-informed Federated Learning for Robust and Explainable Radiology AI" with file number NGF.1609.241.009 of the research programme AiNED XS Europa which is (partly) financed by the Dutch Research Council (NWO).

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

## Appendix A. Ordinal Metrics

The Mean Absolute Error (MAE) reflects higher numerical differences between the actual and predicted labels, resulting in higher penalisation of bigger mistakes over smaller mistakes. The error sum is then averaged over all $M$ observations.

$$MAE = \frac{1}{M} \sum_{i=1}^{M} |y_i - \hat{y}_i| \tag{6}$$

The first disadvantage presented by MAE is its dependence on the number arbitrarily assigned to each class. This can be fixed by defining the classes by their indexes on a

confusion matrix, but MAE will still equally penalised "forwards" and "backwards" errors. In ordinal classification problems, where ranking plays a major role, this lack of distinction between errors is a significant flaw.

Kendall's $\tau_b$ (Kendall, 1938) takes into account ranking in the measurement of classification performance, and it is baed on two rank vector $p$ and $q$:

$$\tau_b = \frac{\sum q_{ij} p_{ij}}{\sqrt{\sum q_{ij}^2 \sum p_{ij}^2}}, \tag{7}$$

where $q_i j$ behaves as follows:

$$\begin{cases} q_{ij} = 1, & \text{if } q_i > q_j, \\ q_{ij} = 0, & \text{if } q_i = q_j, \\ q_{ij} = -1, & \text{if } q_i < q_j, \end{cases} \tag{8}$$

and the same is true for $p_{ij}$. Therefore, the value of $\tau_b$ varies between -1 and 1. However, in this metric the only thing that matters is the relation between classes, causing critical loss of information on absolute classification error.

The Uniform Ordinal Classification Index ($A_{uoc}$) (Silva et al., 2018) address the aforementioned shortcomings by combining aspects of classification accuracy and ranking error. It also takes into consideration imbalanced classes and unobserved categories. The $A_{uoc}$ is derived by tracing paths through the confusion matrix from the top-left to the bottom-right (diagonal). Paths are evaluated based on:

- Benefit: rewards large values (correct predictions) along the path.

- Penalty: penalizes deviations from the diagonal based on the distance between the predicted and true classes.

The metric is computed as

$$A_{uoc} = \int_0^1 UOC_\beta^1 d\beta \tag{9}$$

where

$$UOC_\beta^1 = \min \left\{ 1 - \frac{\sum_{y,\hat{y}\in\text{path}}}{N + \sum_{\forall(y,\hat{y})} p(\hat{y}|y)|y - \hat{y}|} + \frac{\beta}{N} \sum_{(y,\hat{y})\in\text{path}} p(\hat{y}|y)|y - \hat{y}| \right\} \tag{10}$$

with $N$ equal to the number of classes and $y$ and $\hat{y}$ respectively the true and predicted label.

## Appendix B. Federated Learning Methodologies

In FL methods multiple devices collect data and a central server coordinates the global learning objective across the network. In particular, the aim is to minimise:

$$\min_{w} f(w) = \sum_{i=1}^{K} p_i F_i(\omega) \tag{11}$$

where $K$ is the number of devices, $p_i \geq 0$, and $\sum_i p_i = 1$. FedAvg (McMahan et al., 2017) sets $pi = \frac{n_i}{n}$, where $n_i$ the number of samples available at each device $i$, and $n = \sum_i n_i$ is the total number of data points.

FedProx (Li et al., 2020) adds a proximal term to the local subproblem to limit the impact of variable local updates. Therefore, the local objective becomes:

$$\min_{w} h(w_i^t; w^t) = F(w_i^t) + \frac{\mu}{2}||w_i^t - w^t||^2 \tag{12}$$

where $w_i^t$ are the weights of the local model of device $i$ and $w^t$ are the weights of the global model at communication round $t$.

MOON (Li et al., 2021) aims to reduce the distance between the representation learnt by the local model and the representation learnt by the global model, and increase the distance between the representation learnt by the local model and the representation learnt by the previous local model. The network has three components: a base encoder, a projection head, and an output layer. The local loss consists of two parts, the standard loss term for the classification task (e.g., Cross-Entropy), $l_{sup}$, and a model-contrastive loss term, $l_{con}$:

$$l_{con} = -\log \frac{\exp(\text{sim}(z, z_{glob})/\tau)}{\exp(\text{sim}(z, z_{glob})/\tau) + \exp(\text{sim}(z, z_{prev})/\tau)} \tag{13}$$

where $z$, $z_{glob}$, and $z_{prev}$ are the representations extracted by the projection head for the local model, the global model and the local model at the previous communication round, and $\tau$ is a temperature parameter.

Therefore, the loss of an input $(x, y)$ is computed by

$$l = l_{sup}(w_i^t; (x, y)) + \mu l_{con}(w_i^t; w_i^{t-1}; w^t; x) \tag{14}$$

where $\mu$ is a hyperparameter that controls the weight of the model-contrastive loss.

FedALA (Zhang et al., 2023) exploits the Adaptive Local Aggregation (ALA) module that element-wisely aggregates the global model and local model to adapt to the local objective, instead of overwriting it:

$$\hat{w}_i^t = w_i^{t-1} + (w^{t-1} - w_i^{t-1}) \odot W_i \tag{15}$$

where $W_i$ represents the aggregating weights. To reduce computation overhead, FedALA can be re-written with the hyperparameter $p$ to control the range of ALA by applying it on $p$ higher layers and overwriting the parameters in the lower layers in FedAvg fashion:

$$\hat{w}_i^t = w_i^{t-1} + (w^{t-1} - w_i^{t-1}) \odot [\mathbb{K}^{|w_i|-p}; W_i^p] \tag{16}$$

where $|w_i|$ is the number of layers in $w_i^{t-1}$ and $\mathbb{K}^{|w_i|-p}$ has the same shape of the lower layers in $w_i^{t-1}$. The elements in $\mathbb{K}^{|w_i|-p}$ are ones. The values in $W_i^p$ are initialised to ones and then $W_i^p$ is learnt based on the old $W_i^p$ in each iteration. To reduce computation

overhead, $s\%$ of $n_i$ in communication round $t$ is randomly sampled and denoted as $n_i^{s,t}$. Device $i$ trains $W_i^p$ through the gradient-based learning method:

$$W_i^p \leftarrow W_i^p - \eta \nabla_{W_i^p} \mathcal{L}(\hat{w}_i^t, n_i^{s,t}; w^{t-1}) \tag{17}$$

where $\eta$ is the learning rate for weight learning, which we set to 1.

## Appendix C. Hyperparameters Fine-tuning

To determine the hyperparameters for the various approaches, we conduct a grid-search over different values for each hyperparameter, using the train/validation split defined by $(\alpha, p) = (0.90, 0.80)$ with seed = 0. All Federated Learning (FL) methods are trained with Cross-Entropy (CE) loss. The grid-search values and the final selected hyperparameter values are summarized below:

- Communication rounds and local updates: $[(100, 1), (20, 5)]$.

- Learning rate: $[1e-4, 1e-5]$.

FedAvg and FedALA are trained for 20 communication rounds with 5 local updates, while FedProx and MOON are trained for 100 with 1 local update. All methods use a learning rate of $1e-5$, except FedProx, which uses a learning rate of $1e-4$.

For FL-method-specific hyperparameters, the following ranges were explored (notation is consistent with the original implementation papers):

- FedProx: $\mu = [0.001, 0.01, 0.1, 0.5, 1.0]$, with the final selection $\mu = 0.5$.

- MOON: $\mu = [0.1, 5.0, 10.0]$ and $\tau = [0.5, 1.0]$, with the final selection $\mu = 5.0$, $\tau = 0.5$.

- FedALA: $p = 2$ and $s = [80, 100]$, with the final selection $s = 80$.

The hyperparameter $\eta$ of the Binomial Cross-Entropy (BCE), discussed in Subsection 2.1, is set to 1, which is the default value provided by the `dlordinal` library (Bérchez-Moreno et al., 2024), used in the implementation.

Notably, the optimal values in the highest heterogeneity setting may not be ideal for lower heterogeneity scenarios. In lower heterogeneity settings, client data are more balanced and the degree of divergence between local models is reduced, which can change the sensitivity to these hyperparameters. In the following paragraphs, we analyse more in depth how this might have affected each FL approach.

- **FedAvg**: the hyperparameters tuned include standard training parameters such as the number of communication rounds, local updates, and learning rate. Although FedAvg does not explicitly incorporate regularization or adaptation mechanisms targeting heterogeneity, the optimal number of local updates could still depend on the heterogeneity setting. For instance, under extreme heterogeneity, where clients might contain predominantly outer classes that are easier to separate, fewer local updates might suffice to avoid client drift. However, this choice might be suboptimal in more

homogeneous scenarios, where a greater number of local updates could enhance learning without substantially increasing drift. Therefore, while FedAvg's hyperparameters may be relatively less sensitive to heterogeneity compared to other approaches, the optimal configuration of local updates could still vary depending on the level of data heterogeneity.

- **FedProx**: the $\mu$ parameter is critical for controlling the penalty on the deviation of local models from the global model. In the highest heterogeneity setting, a higher $\mu$ can effectively limit client drift. However, when applied to lower heterogeneity scenarios where local models are inherently closer in distribution, the same high values of $\mu$ can overly constrain local updates. This may slow convergence or limit the capacity of local models to learn effectively, leading to suboptimal performance in less heterogeneous settings.

  An alternative worth exploring is the adaptive adjustment of $\mu$. In the original Fed-Prox paper (Li et al., 2020), the authors test on synthetic data a heuristic where $\mu$ is initialized differently based on data distribution (starting at $\mu = 1$ for IID data and $\mu = 0$ for non-IID data) and then adjusted by $\delta = 0.1$ every 5 rounds, decreasing when the loss consistently decreases and increasing when the loss rises. Despite this proposal, most studies that employ FedProx as a baseline tend to fine-tune a fixed value of $\mu$ rather than using the adaptive approach. For instance, in the MOON paper (Li et al., 2021), the authors fine-tune $\mu$ for each dataset but keep it fixed across different heterogeneity settings. In future work, we plan to explore this adaptive approach further by also fine-tuning the adjustment factor $\delta$. To better understand the impact of $\mu$ on our experiments, we have re-run the hyperparameter tuning for the middle heterogeneity setting ($\alpha = 2.00, p = 0.90$). The newly selected values are the following:

  – communication rounds: 100.

  – local updates: 1.

  – learning rate: $1e - 5$.

  – $\mu$: 0.1.

  Results are reported in Table C. Due to time constraints, we were not able to re-run the fine-tuning for the remaining heterogeneity settings. However, these additional results further confirm the heavy dependence of FedProx on $\mu$, regardless of the loss function used for training.

- **MOON**: the hyperparameters fine-tuned are $\mu$ (contrastive loss weight) and $\tau$ (temperature parameter). In the original MOON paper (Li et al., 2021), $\tau$ is fixed at 0.5, which our tuning also confirmed as optimal. The original experiments were conducted on CIFAR-10, CIFAR-100 and Tiny-ImageNet, where data were partitioned among 10 clients using a Dirichlet distribution with $\alpha = 0.5$, and $\mu$ was tuned for each dataset under this setting. Subsequently, robustness analysis was performed on CIFAR-100 by varying the heterogeneity ($\alpha = 0.1$ and $\alpha = 5.0$), without re-tuning $\mu$. Despite using a fixed $\mu$ across different heterogeneity levels, MOON consistently outperformed

Table 1: Performance comparison of FedProx using CE, BCE, and OE under intermediate heterogeneity conditions ($\alpha = 2.00, p = 0.90$). The upper section reports metrics ($A_{uoc}$, MAE, Kendall's $\tau$, Balanced Accuracy) for the newly selected hyperparameters after re-tuning (communication rounds $r = 100$, local updates $lu = 1$, learning rate lr $= 1 \times 10^{-5}$, proximal term $\mu = 0.1$). The lower section shows results for the previously selected hyperparameters (communication rounds $r = 100$, local updates $lu = 1$, learning rate $1 \times 10^{-4}$, proximal term $\mu = 0.5$). Results are averaged across three runs and include standard deviations.

| (2.00,0.90) | Auoc ↓ | MAE ↓ | Kendall's tau ↑ | Balanced Accuracy ↑ |
|---|---|---|---|---|
| | | r=100, lu=1, lr=1e-5, mu=0.1 | | |
| CE | 0.7310±0.0251 | 0.8399±0.0465 | 0.7202±0.0068 | 0.3850±0.0470 |
| BCE | 0.7199±0.0254 | 0.7829±0.0693 | 0.7374±0.0041 | 0.3782±0.0450 |
| OE | 0.7428±0.0218 | 0.8594±0.0671 | 0.7122±0.0210 | 0.3645±0.0178 |
| | | r=100, lu=1, lr=1e-4, mu=0.5 | | |
| CE | 0.7789±0.0346 | 1.0200±0.1312 | 0.7027±0.0263 | 0.3266±0.0367 |
| BCE | 0.7739±0.0182 | 0.9549±0.0737 | 0.6828±0.0194 | 0.3164±0.0235 |
| OE | 0.7628±0.0114 | 0.9531±0.0602 | 0.7194±0.0208 | 0.3377±0.0144 |

the baselines. Their choice of tuning at intermediate heterogeneity could be argued as balanced for evaluating robustness across extremes. In contrast, our work explicitly focuses on understanding model behaviour at extreme heterogeneity, motivating our decision to tune hyperparameters specifically for the highest heterogeneity setting. It is worth noting, however, that while these hyperparameter settings effectively mitigate misalignment in extreme data heterogeneity, in lower heterogeneity settings they might enforce an unnecessarily strong alignment.

- **FedALA**: The hyperparameters to be fine-tuned for FedALA are $p$ (the number of layers used for adaptive aggregation) and $s$ (the sample proportion applied during aggregation). In the original FedALA paper (Zhang et al., 2023), the authors evaluated the method on four computer vision datasets and one natural language processing dataset. They performed the tuning of $p$ and $s$ on the Tiny-ImageNet dataset, which was partitioned across 20 clients using a Dirichlet distribution with $\alpha = 0.1$. Their results indicated that while $s = 100$ yielded slightly better performance than $s = 80$, the difference was negligible; thus, $s = 80$ was chosen for all datasets and experiments to balance performance with computational cost. Additionally, $p = 2$ was found to provide the best results. In our tuning process, we similarly focused on fine-tuning $s$.

Furthermore, the original authors conducted extra experiments to assess robustness under varying heterogeneity levels ($\alpha \in \{0.01, 0.5, 1\}$), yet they did not re-tune the hyperparameters for each different level. Given that the performance variations across different values of $s$ and $p$ were minimal, they concluded that the effects of these hyperparameters are largely negligible in different heterogeneity settings. Nonetheless, it is worth noting that since this analysis was carried out on only one dataset, the adaptive mechanism controlled by these hyperparameters might result in over-compensation in settings with lower heterogeneity.

Table 2: Comparison of the centralised baseline and FedAvg in the lowest heterogeneity scenario ($\alpha = 10.00, p = 0.99$). Results are reported for $A_{uoc}$, MAE, Kendall's $\tau$, and Balanced Accuracy. Results are averaged and reported with the standard deviation across three runs.

| | **Auoc ↓** | **MAE ↓** | **Kendall's $\tau$ ↑** | **Balanced Accuracy ↑** |
|---|---|---|---|---|
| | **Centralised** | | | |
| CE | 0.7403±0.0129 | 0.8575±0.0156 | 0.7051±0.0129 | 0.3892±0.0184 |
| BCE | 0.7077±0.0123 | 0.7398±0.0233 | 0.7347±0.0138 | 0.3960±0.0084 |
| OE | 0.7319±0.0108 | 0.8323±0.0260 | 0.7217±0.0027 | 0.3895±0.0126 |
| | **FedAvg (10.00,0.99)** | | | |
| CE | 0.7366±0.0088 | 0.8301±0.0302 | 0.7018±0.0088 | 0.3891±0.0131 |
| BCE | 0.7257±0.0158 | 0.7578±0.0274 | 0.7251±0.0140 | 0.3831±0.0335 |
| OE | 0.7231±0.0177 | 0.7978±0.0395 | 0.7361±0.0061 | 0.4017±0.0221 |

Overall, current methods for handling client heterogeneity in federated learning rely heavily on hyperparameter tuning. While this approach performs well in controlled experiments where heterogeneity levels are known, it may not be robust in real-world settings where these levels are unpredictable. Future research should focus on developing adaptive methods that dynamically adjust to varying heterogeneity, while also preserving ordinal relationships when necessary.

## Appendix D. Additional Results

### D.1. Centralised Baseline

We present the centralised baseline results for CE, BCE, and OE, trained using data splits from the lowest heterogeneity scenario (($\alpha = 10.00, p = 0.99$)), which best approximates a centralised setting. In Table 2, these results are compared to those from FedAvg trained on the same splits. Notably, FedAvg performs similarly to, and in some cases even outperforms, the centralised baseline.

### D.2. Runtime analysis

We performed a runtime analysis to evaluate computational costs across methods. Specifically, we measured epoch and aggregation times for each FL approach. The experiments were conducted for 10 communication rounds and 1 local update on a NVIDIA GPU A6000, trained on the train/validation split defined by $(\alpha, p) = (0.90, 0.80)$ with seed = 0. Epoch runtimes for each client are shown in Table 3, while the average aggregation times per method are reported in Table 4. We observe that employing ordinal appraoches does not affect runtime. However, differences arise due to the federated learning methods themselves. In particular, MOON shows higher epoch runtimes compared to FedAvg and FedProx because of the additional computations required for the contrastive loss. In terms of aggregation times, MOON and FedProx perform the same aggregation procedure as FedAvg, resulting in similar aggregation durations across these three methods. Conversely, FedALA

Table 3: Epoch runtimes (in seconds) for each approach on each client (mean ± standard deviation). The bolded row for each approach indicates the average epoch time across all five clients, with the standard deviation computed across those clients.

|  | **FedAvg** | **FedProx** | **MOON** | **FedALA** |
|---|---|---|---|---|
| **CE** | **20.6333±3.4892** | **20.9081±3.5325** | **35.8841±6.4658** | **19.9587±3.4728** |
| client 0 | 20.5991±0.6149 | 21.0684±0.3874 | 36.0840±0.3584 | 21.0252±3.5868 |
| client 1 | 16.4326±0.4382 | 16.6843±0.3276 | 28.0037±0.1635 | 15.7149±0.6115 |
| client 2 | 24.9204±0.6542 | 25.2264±0.5590 | 43.4734±0.1482 | 23.8610±1.1974 |
| client 3 | 18.0903±0.6021 | 18.1954±0.3164 | 31.0572±0.0847 | 16.9883±0.7097 |
| client 4 | 23.1241±0.4612 | 23.3661±0.3780 | 40.8023±0.0913 | 22.2041±0.9153 |
| **BCE** | **20.9500±3.6083** | **21.0632±0.0797** | **35.8950±6.4504** | **18.6942±3.2817** |
| client 0 | 20.8436±0.3746 | 21.1456±0.5905 | 36.0957±0.0797 | 18.6777±0.4864 |
| client 1 | 16.7477±0.5032 | 16.8931±0.4697 | 28.0777±0.1016 | 14.8505±0.3069 |
| client 2 | 25.4342±0.6218 | 25.2165±0.4802 | 43.4598±0.1218 | 22.6341±0.5376 |
| client 3 | 18.1897±0.3545 | 18.4162±0.3665 | 31.0191±0.1220 | 16.1266±0.5151 |
| client 4 | 23.5347±0.6556 | 23.6448±0.4585 | 40.8225±0.0909 | 21.1819±1.0033 |
| **OE** | **20.9935±3.4795** | **20.8317±0.1756** | **35.8276±6.4546** | **18.6895±3.1515** |
| client 0 | 21.0068±0.3855 | 21.0377±0.6390 | 36.0054±0.1137 | 18.6975±0.6909 |
| client 1 | 17.0115±0.2657 | 16.4822±0.4228 | 28.0091±0.0775 | 14.9748±0.3864 |
| client 2 | 25.3031±0.5451 | 25.0936±0.4700 | 43.3911±0.1172 | 22.5083±0.4264 |
| client 3 | 18.1830±0.3451 | 18.1438±0.2320 | 30.9532±0.1075 | 16.2525±0.3446 |
| client 4 | 23.4632±0.5094 | 23.4011±0.6625 | 40.7793±0.0934 | 21.0145±0.4071 |

Table 4: Average aggregation runtimes (in seconds) reported with standard deviation across communication rounds.

|  | **FedAvg** | **FedProx** | **MOON** | **FedALA** |
|---|---|---|---|---|
| CE | 0.0205±0.0020 | 0.0201±0.0004 | 0.0205±0.0003 | 384.9968±595.5023 |
| BCE | 0.0198±0.0001 | 0.0201±0.0002 | 0.0205±0.0002 | 367.4667±557.1648 |
| OE | 0.0199±0.0003 | 0.0202±0.0003 | 0.0204±0.0001 | 367.1724±551.8641 |

exhibits higher aggregation times with greater variability. This is explained by FedALA's adaptive local aggregation (ALA) module, which dynamically learns weights to combine local and global model parameters. The runtime variance occurs because, in communication rounds where local and global models differ substantially, the ALA module requires more iterations to converge, whereas in rounds with closer models, convergence is faster.

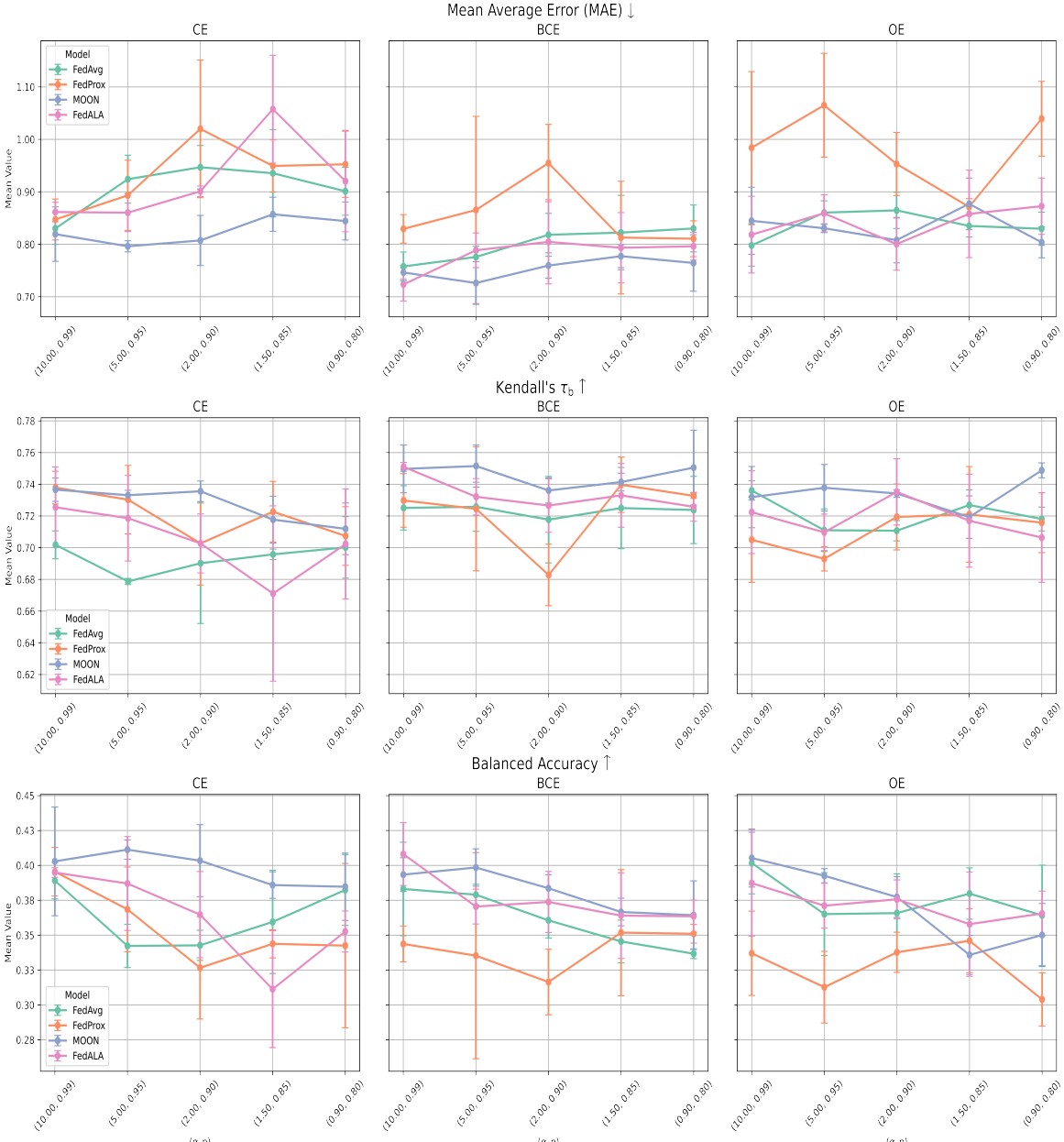

Figure 4: Performance trends for the four Federated Learning methods (FedAvg, FedProx, MOON, FedALA) under increasing data heterogeneity (x-axis: different $\alpha$ and $p$ values), comparing approaches: Cross-Entropy (CE), Binomial Cross-Entropy (BCE), and Ordinal Encoding (OE). The top row shows the Mean Average Error (MAE), the middle row shows Kendall's $\tau_b$, while the bottom row shows Balanced Accuracy, with error bars representing the standard deviation across three runs.

Table 5: Comparison of Uniform Ordinal Classification Index ($A_{uoc}$) for the four Federated Learning methods (FedAvg, FedProx, MOON, FedALA) under increasing data heterogeneity (different $\alpha$ and $p$ values), comparing approaches: Cross-Entropy (CE), Binomial Cross-Entropy (BCE), and Ordinal Encoding (OE). Results are averaged and reported with the standard deviation across three runs.

| (alpha, p) | Auoc ↓ | | | |
|---|---|---|---|---|
| **(10.00, 0.99)** | FedAvg | FedProx | MOON | FedALA |
| CE | 0.7366±0.0088 | 0.7281±0.0126 | 0.7206±0.0238 | 0.7341±0.0031 |
| BCE | 0.7257±0.0158 | 0.7372±0.0077 | 0.7062±0.0101 | 0.7018±0.0161 |
| OE | 0.7231±0.0177 | 0.7733±0.0305 | 0.7261±0.0209 | 0.7340±0.0326 |
| **(5.00, 0.95)** | FedAvg | FedProx | MOON | FedALA |
| CE | 0.7650±0.0062 | 0.7514±0.0234 | 0.7130±0.0044 | 0.7390±0.0175 |
| BCE | 0.7421±0.0179 | 0.7470±0.0597 | 0.7033±0.0180 | 0.7328±0.0281 |
| OE | 0.7486±0.0090 | 0.7897±0.0244 | 0.7275±0.0012 | 0.7440±0.0130 |
| **(2.00, 0.90)** | FedAvg | FedProx | MOON | FedALA |
| CE | 0.7650±0.0110 | 0.7789±0.0346 | 0.7224±0.0174 | 0.7529±0.0025 |
| BCE | 0.7421±0.0179 | 0.7739±0.0182 | 0.7165±0.0112 | 0.7325±0.0281 |
| OE | 0.7440±0.0180 | 0.7628±0.0114 | 0.7264±0.0133 | 0.7267±0.0204 |
| **(1.50, 0.85)** | FedAvg | FedProx | MOON | FedALA |
| CE | 0.7605±0.0192 | 0.7617±0.0094 | 0.7348±0.0092 | 0.7920±0.0277 |
| BCE | 0.7426±0.0251 | 0.7286±0.0374 | 0.7266±0.0028 | 0.7321±0.0253 |
| OE | 0.7315±0.0078 | 0.7519±0.0060 | 0.7488±0.0102 | 0.7419±0.0303 |
| **(0.90, 0.80)** | FedAvg | FedProx | MOON | FedALA |
| CE | 0.7469±0.0053 | 0.7647±0.0349 | 0.7373±0.0125 | 0.7629±0.0172 |
| BCE | 0.7502±0.0147 | 0.7344±0.0080 | 0.7160±0.0223 | 0.7375±0.0103 |
| OE | 0.7386±0.0155 | 0.7952±0.0085 | 0.7320±0.0028 | 0.7476±0.0069 |

Table 6: Comparison of Mean Average Error (MAE) for the four Federated Learning methods (FedAvg, FedProx, MOON, FedALA) under increasing data heterogeneity (different $\alpha$ and $p$ values), comparing approaches: Cross-Entropy (CE), Binomial Cross-Entropy (BCE), and Ordinal Encoding (OE). Results are averaged and reported with the standard deviation across three runs.

| (alpha, p) | MAE ↓ | | | |
|---|---|---|---|---|
| **(10.00, 0.99)** | FedAvg | FedProx | MOON | FedALA |
| CE | 0.8301±0.0302 | 0.8473±0.0390 | 0.8194±0.0519 | 0.8615±0.0186 |
| BCE | 0.7578±0.0274 | 0.8294±0.0272 | 0.7463±0.0131 | 0.7234±0.0316 |
| OE | 0.7978±0.0395 | 0.9840±0.1452 | 0.8448±0.0639 | 0.8183±0.0731 |
| **(5.00, 0.95)** | FedAvg | FedProx | MOON | FedALA |
| CE | 0.9241±0.0457 | 0.8933±0.0670 | 0.7962±0.0106 | 0.8603±0.0361 |
| BCE | 0.7758±0.0204 | 0.8654±0.1785 | 0.7262±0.0407 | 0.7886±0.0331 |
| OE | 0.8604±0.0225 | 1.0650±0.0991 | 0.8306±0.0084 | 0.8591±0.0355 |
| **(2.00, 0.90)** | FedAvg | FedProx | MOON | FedALA |
| CE | 0.9469±0.0415 | 1.0200±0.1312 | 0.8073±0.0477 | 0.9010±0.0101 |
| BCE | 0.8181±0.0409 | 0.9549±0.0737 | 0.7595±0.0242 | 0.8046±0.0802 |
| OE | 0.8647±0.0345 | 0.9531±0.0602 | 0.8079±0.0433 | 0.8003±0.0498 |
| **(1.50, 0.85)** | FedAvg | FedProx | MOON | FedALA |
| CE | 0.9355±0.0832 | 0.9494±0.0500 | 0.8572±0.0325 | 1.0576±0.1028 |
| BCE | 0.8223±0.0709 | 0.8128±0.1074 | 0.7773±0.0219 | 0.7934±0.0670 |
| OE | 0.8350±0.0211 | 0.8713±0.0158 | 0.8771±0.0492 | 0.8580±0.0834 |
| **(0.90, 0.80)** | FedAvg | FedProx | MOON | FedALA |
| CE | 0.9011±0.0457 | 0.9525±0.0631 | 0.8443±0.0362 | 0.9205±0.0966 |
| BCE | 0.8302±0.0447 | 0.8106±0.0342 | 0.7644±0.0539 | 0.7962±0.0264 |
| OE | 0.8296±0.0317 | 1.0394±0.0714 | 0.8038±0.0296 | 0.8726±0.0537 |

Table 7: Comparison of Kendall's $\tau_b$ for the four Federated Learning methods (FedAvg, FedProx, MOON, FedALA) under increasing data heterogeneity (different $\alpha$ and $p$ values), comparing approaches: Cross-Entropy (CE), Binomial Cross-Entropy (BCE), and Ordinal Encoding (OE). Results are averaged and reported with the standard deviation across three runs.

| (alpha, p) | Kendall's tau ↑ | | | |
|---|---|---|---|---|
| **(10.00, 0.99)** | FedAvg | FedProx | MOON | FedALA |
| CE | 0.7018±0.0088 | 0.7379±0.0103 | 0.7366±0.0074 | 0.7255±0.0254 |
| BCE | 0.7251±0.0140 | 0.7298±0.0169 | 0.7497±0.0151 | 0.7510±0.0028 |
| OE | 0.7361±0.0061 | 0.7050±0.0269 | 0.7319±0.0193 | 0.7224±0.0261 |
| **(5.00, 0.95)** | FedAvg | FedProx | MOON | FedALA |
| CE | 0.6787±0.0019 | 0.7303±0.0216 | 0.7331±0.0031 | 0.7186±0.0270 |
| BCE | 0.7258±0.0155 | 0.7245±0.0391 | 0.7515±0.0133 | 0.7321±0.0115 |
| OE | 0.7110±0.0134 | 0.6930±0.0077 | 0.7378±0.0147 | 0.7097±0.0115 |
| **(2.00, 0.90)** | FedAvg | FedProx | MOON | FedALA |
| CE | 0.6902±0.0380 | 0.7027±0.0263 | 0.7356±0.0065 | 0.7027±0.0186 |
| BCE | 0.7177±0.0273 | 0.6828±0.0194 | 0.7361±0.0080 | 0.7266±0.0169 |
| OE | 0.7107±0.0064 | 0.7194±0.0208 | 0.7342±0.0024 | 0.7352±0.0209 |
| **(1.50, 0.85)** | FedAvg | FedProx | MOON | FedALA |
| CE | 0.6958±0.0033 | 0.7227±0.0191 | 0.7177±0.0148 | 0.6711±0.0553 |
| BCE | 0.7250±0.0256 | 0.7397±0.0175 | 0.7414±0.0056 | 0.7330±0.0201 |
| OE | 0.7269±0.0096 | 0.7210±0.0302 | 0.7193±0.0134 | 0.7170±0.0293 |
| **(0.90, 0.80)** | FedAvg | FedProx | MOON | FedALA |
| CE | 0.7002±0.0194 | 0.7074±0.0185 | 0.7119±0.0163 | 0.7023±0.0347 |
| BCE | 0.7238±0.0212 | 0.7327±0.0016 | 0.7505±0.0236 | 0.7258±0.0091 |
| OE | 0.7181±0.0075 | 0.7158±0.0189 | 0.7488±0.0047 | 0.7064±0.0282 |

Table 8: Comparison of Balanced Accuracy for the four Federated Learning methods (FedAvg, FedProx, MOON, FedALA) under increasing data heterogeneity (different $\alpha$ and $p$ values), comparing approaches: Cross-Entropy (CE), Binomial Cross-Entropy (BCE), and Ordinal Encoding (OE). Results are averaged and reported with the standard deviation across three runs.

| (alpha, p) | Balanced accuracy ↑ | | | |
|---|---|---|---|---|
| **(10.00, 0.99)** | FedAvg | FedProx | MOON | FedALA |
| CE | 0.3891±0.0131 | 0.3955±0.0174 | 0.4029±0.0390 | 0.3949±0.0035 |
| BCE | 0.3831±0.0335 | 0.3437±0.0129 | 0.3934±0.0121 | 0.4081±0.0225 |
| OE | 0.4017±0.0221 | 0.3370±0.0302 | 0.4053±0.0208 | 0.3873±0.0380 |
| **(5.00, 0.95)** | FedAvg | FedProx | MOON | FedALA |
| CE | 0.3423±0.0155 | 0.3685±0.0304 | 0.4113±0.0069 | 0.3870±0.0337 |
| BCE | 0.3790±0.0074 | 0.3353±0.0739 | 0.3985±0.0134 | 0.3705±0.0125 |
| OE | 0.3651±0.0297 | 0.3127±0.0257 | 0.3926±0.0051 | 0.3711±0.0161 |
| **(2.00, 0.90)** | FedAvg | FedProx | MOON | FedALA |
| CE | 0.3427±0.0108 | 0.3266±0.0367 | 0.4034±0.0258 | 0.3647±0.0309 |
| BCE | 0.3606±0.0127 | 0.3164±0.0235 | 0.3836±0.0098 | 0.3738±0.0219 |
| OE | 0.3658±0.0282 | 0.3377±0.0144 | 0.3773±0.0146 | 0.3757±0.0139 |
| **(1.50, 0.85)** | FedAvg | FedProx | MOON | FedALA |
| CE | 0.3595±0.0370 | 0.3438±0.0101 | 0.3859±0.0095 | 0.3113±0.0419 |
| BCE | 0.3455±0.0153 | 0.3518±0.0452 | 0.3666±0.0099 | 0.3640±0.0307 |
| OE | 0.3798±0.0184 | 0.3460±0.0229 | 0.3358±0.0140 | 0.3579±0.0374 |
| **(0.90, 0.80)** | FedAvg | FedProx | MOON | FedALA |
| CE | 0.3824±0.0254 | 0.3425±0.0589 | 0.3847±0.0241 | 0.3526±0.0147 |
| BCE | 0.3367±0.0035 | 0.3510±0.0067 | 0.3643±0.0246 | 0.3635±0.0117 |
| OE | 0.3642±0.0360 | 0.3039±0.0191 | 0.3501±0.0226 | 0.3655±0.0161 |

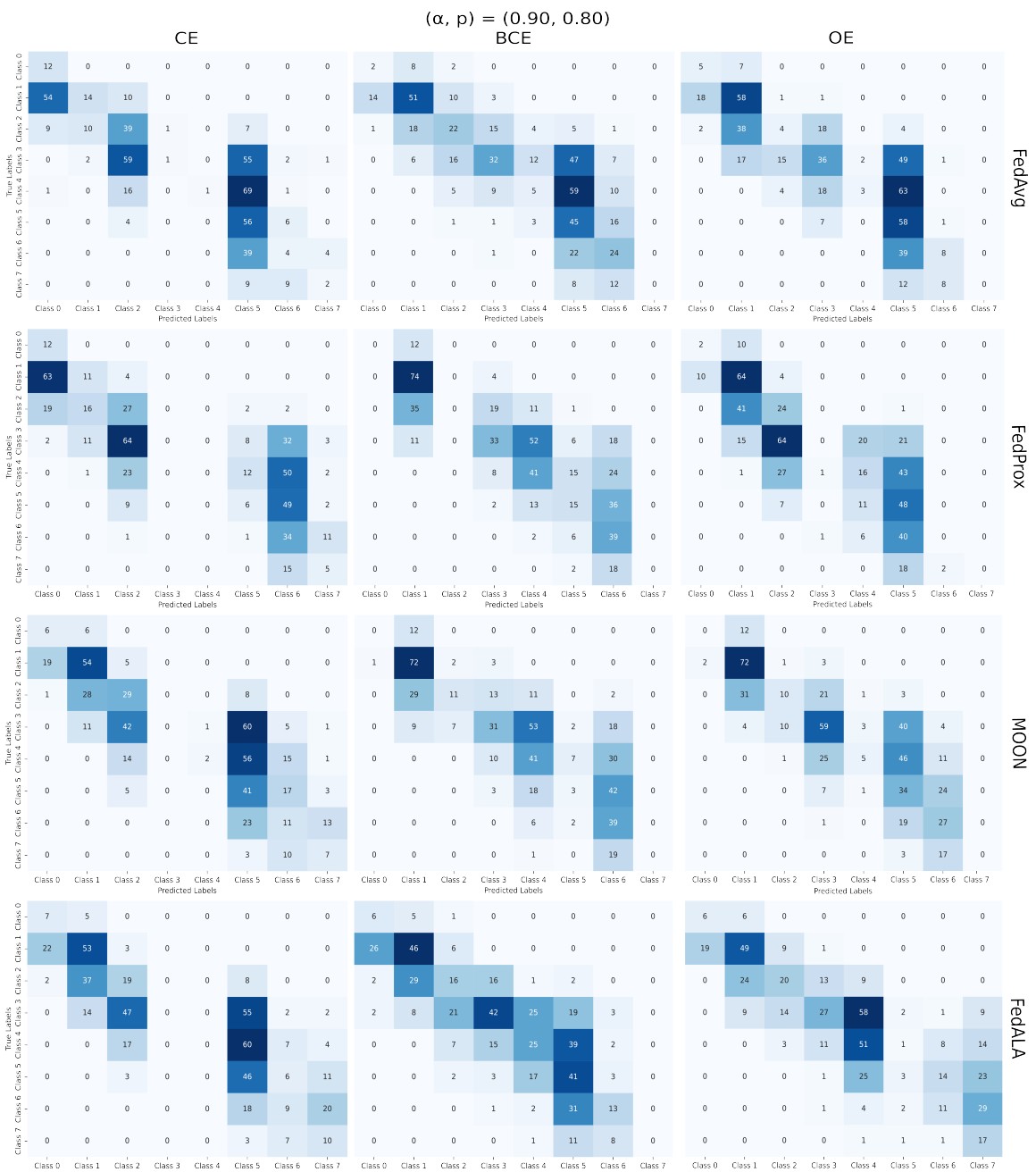

Figure 5: Confusion matrices for the highest heterogeneity setting, with $(\alpha, p) = (0.90, 0.80)$ train/validation split (seed = 0) for the Federated Learning methods (FedAvg, FedProx, MOON, FedALA) with Cross-Entropy (CE), Binomial Cross-Entropy (BCE), and Ordinal Encoding.

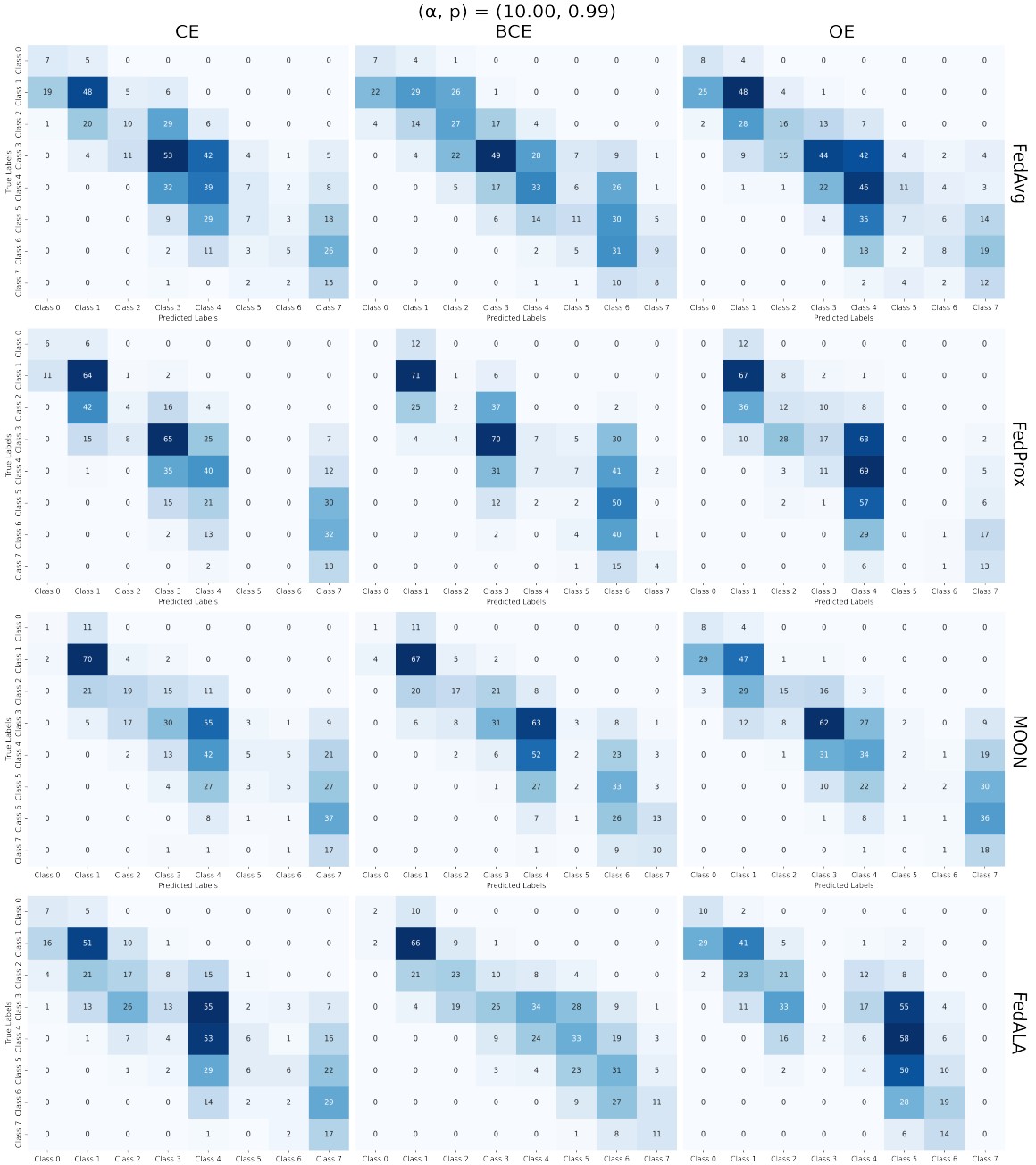

Figure 6: Confusion matrices for the lowest heterogeneity setting, with $(\alpha, p) = (10.00, 0.99)$ train/validation split (seed $= 0$) for the Federated Learning methods (FedAvg, FedProx, MOON, FedALA) with Cross-Entropy (CE), Binomial Cross-Entropy (BCE), and Ordinal Encoding.

