# OpenReview forum: "Understanding the Impact of Client Heterogeneity on Ordinal Classification in Federated Medical Image Analysis"
_MIDL.io/2025/Conference — MIDL 2025 Poster_

### Official Review · Reviewer_xAPJ · 2025-02-15

**Confidence:** 4
**Preliminary Rating:** 4
**Final Rating:** 4

**Summary:**

This paper investigates the impact of client data heterogeneity on Ordinal Classification within Federated Learning (FL) for medical image analysis. The authors introduce Federated Ordinal Learning (FOL) and evaluate its performance under different levels of heterogeneity. They compare Cross-Entropy (CE) with two ordinal classification methods: Binomial Cross-Entropy (BCE) and Ordinal Encoding (OE). The experiments are performed on the CSAW-M mammography dataset, simulating varying levels of data heterogeneity using a Bernoulli-Dirichlet sampling strategy. Results demonstrate that ordinal methods (BCE, OE) outperform standard classification (CE) under heterogeneous conditions, but performance still degrades significantly as heterogeneity increases.

**Strengths:**

1. The paper integration of Ordinal Classification into FL for medical image analysis, it is a significant step for privacy-preserving healthcare AI.
2. The study rigorously tests multiple ordinal methods across four FL algorithms, providing valuable insights into their behavior under different heterogeneity levels.
3. The authors use a controlled heterogeneity simulation on a benchmark dataset (CSAW-M), ensuring a clear comparison of methods.

**Weaknesses:**

1. The study only uses one dataset (CSAW-M) from a single source, limiting the generalizability to other imaging modalities.
2. FedProx underperforms due to poor μ tuning, which was only optimized for the highest heterogeneity scenario, suggesting a lack of thorough hyperparameter exploration.
3. The paper does not provide any model interpretability analysis to explain the clinical relevance of ordinal predictions.

**Detailed Comments:**

1. Future work should test on multiple datasets
2. The paper only enable 5 clients to do FL, it is not enough, usualy the experiment is set up 100+ client to proof validation.

**Justification Of The Final Rating:**

The paper makes a valuable contribution by integrating ordinal classification into federated learning for medical image analysis, addressing data heterogeneity. While limitations exist, such as the use of a single dataset and a small number of clients, the authors provide reasonable justifications and acknowledge these as future research directions.

**Justification Of The Preliminary Rating:**

This paper introduces Ordinal Classification into Federated Learning (FL) for medical image analysis, addressing data heterogeneity in privacy-preserving AI. The authors compare four FL methods (FedAvg, FedProx, MOON, FedALA) using two ordinal approaches (BCE and OE) on the CSAW-M mammography dataset, showing that ordinal methods outperform standard classification under heterogeneous conditions. FedALA performs best, particularly when handling missing classes. The study is well-executed with rigorous experiments and insightful analysis.

**Questions To Address In The Rebuttal:**

1.  The study only uses the CSAW-M dataset from a single institution. Can the authors provide additional experiments or discuss how their approach would perform on other datasets or imaging modalities
2.  The performance of FedProx is highly dependent on μ, which was only fine-tuned for the highest heterogeneity scenario. Can the authors provide additional results with hyperparameter sweeps across multiple heterogeneity settings or justify why they did not tune for intermediate scenarios?
3. Can author can extend the experiment to 100+ client and evaluate the results.

---

> ### Author Response · Authors · 2025-03-07
>
> We thank the reviewer for the insightful suggestions. We have partially addressed their remarks in the main rebuttal comment. Below, we address the remaining ones.
>
> ### Limited number of clients
>
> In the medical imaging literature, FL studies typically involve only a handful of clients, often ranging from 2 to 5, due to practical constraints on multi-center data collection. Many works also simulate non-IID scenarios by splitting a single dataset into a small number of clients, as noted by Guan et al [1]. Over the last five years, as FL frameworks have matured, there has been a gradual increase in the number of participating clients, although most medical imaging studies still include fewer than a dozen on average [2][3]. Our decision to use five clients is thus in line with common practice.
>
> In this study, given that we only had 10,020 images, partitioning them into 100+ clients, combined with various heterogeneity settings, would have led to extreme data scarcity at each client. This would introduce an additional confounder related to missing or under-represented classes, which goes beyond our current focus. Nevertheless, we acknowledge that increasing the number of clients could enhance the generalizability of our findings, as it would allow us to investigate the robustness of ordinal approaches under both missing or under-represented classes and data scarcity. Unfortunately, due to the limited timeframe for this rebuttal, we cannot provide results for larger federations at present, but we plan to explore this in future work. We have also noted this limitation in the Conclusions Section.
>
> [1] Hao Guan, Pew-Thian Yap, Andrea Bozoki, and Mingxia Liu. Federated learning for medical image analysis: A survey. Pattern Recognition, page 110424, 2024.
> [2] Nannan Wu, Li Yu, Xin Yang, Kwang-Ting Cheng, and Zengqiang Yan. Fediic: Towards robust federated learning for class-imbalanced medical image classification. In International Conference on Medical Image Computing and Computer-Assisted Intervention,
> pages 692–702. Springer, 2023.
> [3] Yangyang Xiang, Nannan Wu, Li Yu, Xin Yang, Kwang-Ting Cheng, and Zengqiang Yan. Fedia: Federated medical image segmentation with heterogeneous annotation completeness. In International Conference on Medical Image Computing and Computer-Assisted Intervention, pages 373–382. Springer, 2024.
>
> ### Additional comments on the results on only one dataset
>
> We have already addressed this in the comment to reviewer v6qC, but for convenience we repeat it here as well.
> As this is the first evaluation of ordinal approaches in heterogeneous federated learning, we began our analysis using a dataset specifically designed as a benchmark for ordinal classification. The CSAW-M dataset not only was curated with ordinality in mind, but also offers additional advantages. All images originate from a single centre and are annotated using a consistent protocol, which minimizes potential confounding factors due to distribution shifts or variations in annotation strategies. Moreover, the labels in CSAW-M were deliberately designed to reflect ordinal relationships, providing a more robust basis for testing ordinal methods compared to naturally ordinal tasks, such as breast cancer classification using BI-RADS, where the labels are often noisy.
>
> Our choice of the CSAW-M dataset, combined with the Bernoulli-Dirichlet sampling strategy, allowed us to isolate the effects of imbalanced class distributions and missing classes. Additionally, the federated learning methods and ordinal strategies we investigated have been validated across diverse contexts in the literature, suggesting that their behaviour under heterogeneity is not specific to a single dataset. While we recognize the value of testing on multiple datasets, CSAW-M serves as a controlled and reliable benchmark for ordinal classification in medical imaging.
>
> Building on these results, we plan to extend our analysis to additional datasets and tasks. Due to the limited time available for the rebuttal, we are unable to provide these additional results at this stage. We have acknowledged this limitation in the Conclusions Section and have outlined the expansion of our analysis to other datasets as a direction for future work.

---

> ### Comment · Area_Chair_irnW · 2025-03-13
> **Please update final rating**
>
> Dear Reviewer xAPJ, Thank you for reviewing the authors' response/updates. Please update your official final rating and justification by editing your review. The discussion period ends tomorrow, March 14. Thank you!

---

### Official Review · Reviewer_hFHk · 2025-02-21

**Confidence:** 4
**Preliminary Rating:** 4

**Summary:**

This study introduces Federated Ordinal Learning (FOL) to medical imaging, integrating ordinal classification into Federated Learning (FL) for the first time. The research evaluates ordinal methods—Binomial Cross-Entropy (BCE) and Ordinal Encoding (OE)—against the conventional Cross-Entropy (CE) in four FL frameworks: FedAvg, FedProx, MOON, and FedALA. Findings reveal that ordinal approaches enhance robustness to missing classes and class imbalance, outperforming CE.

**Strengths:**

1. Clear and Well-Organized Writing.
The paper is easy to follow, with a strong motivation that clearly highlights the importance of ordinal classification in medical imaging.

2. Well-Motivated BCE & OE Modules.
The introduction of Binomial Cross-Entropy (BCE) and Ordinal Encoding (OE) is intuitive and well-explained. The authors provide solid experimental support showing how these methods preserve ordinal relationships better than standard Cross-Entropy (CE).

3. Comprehensive Experiments on FL & Heterogeneity.
The study systematically evaluates BCE and OE under different levels of data heterogeneity using four FL methods (FedAvg, FedProx, MOON, FedALA). The use of ordinal-specific metrics ensures a thorough analysis.

4. Useful Findings & Research Directions.
The paper identifies key challenges, such as performance degradation under extreme heterogeneity, and proposes future research directions for developing FL strategies that explicitly handle ordinal relationships.

**Weaknesses:**

1. Figure Presentation Could Be Improved. Figures 1 and 3 take up a lot of space but could be more informative and compact.

2. Limited Discussion on Existing Heterogeneous FL Strategies. While the paper highlights data heterogeneity as a major challenge, it does not explore how advanced FL methods designed to address heterogeneity—such as flat minima optimization techniques (e.g., FedSAM [1], FedSoup [2]) or partial fine-tuning strategies (e.g., FedRep [3], FedBABU [4])—could also benefit ordinal classification.

[1] Qu, Zhe, et al. "Generalized federated learning via sharpness aware minimization." International conference on machine learning. PMLR, 2022.
[2] Chen, Minghui, et al. "FedSoup: improving generalization and personalization in federated learning via selective model interpolation." International Conference on Medical Image Computing and Computer-Assisted Intervention. Cham: Springer Nature Switzerland, 2023.
[3] Collins, Liam, et al. "Exploiting shared representations for personalized federated learning." International conference on machine learning. PMLR, 2021.
[4] Oh, Jaehoon, SangMook Kim, and Se-Young Yun. "FedBABU: Toward Enhanced Representation for Federated Image Classification." International Conference on Learning Representations.

**Detailed Comments:**

n/a

**Justification Of The Preliminary Rating:**

This paper makes a valuable contribution to Federated Ordinal Learning (FOL) by systematically integrating ordinal classification into FL for medical imaging, addressing a critical yet underexplored challenge. It is well-written and clearly structured, making the motivation and problem statement easy to follow. The proposed BCE and OE modules are well-explained and experimentally validated, demonstrating their effectiveness in preserving ordinal relationships. The comprehensive experiments across multiple FL methods provide insightful analysis of data heterogeneity, using appropriate ordinal-specific metrics. Moreover, the study identifies key limitations and proposes future research directions, offering a strong foundation for further advancements in heterogeneous ordinal FL.

**Questions To Address In The Rebuttal:**

n/a

---

> ### Author Response · Authors · 2025-03-07
>
> We thank reviewer hFHk for the interesting suggestions. Below we address their remark.
>
> ### Limited discussion on heterogeneous FL methods
>
> In our current study, we focused on integrating ordinal classification into FL using heterogeneity-aware methods (FedProx, MOON, and FedALA) where we could isolate the impact of client heterogeneity on ordinal methods, without having to modify the method itself, but where we could simply plug in the ordinal learning paradigm. We agree that advanced strategies such as the ones suggested by the Reviewer have shown significant promise in addressing heterogeneity. Considering our findings, exploring the interaction between ordinal classification and these advanced heterogeneous FL strategies is an interesting direction for future research. We will extend our analysis to include such methods. We have addressed this future direction in the Conclusions section of our manuscript.

---

> ### Comment · Area_Chair_irnW · 2025-03-13
> **Please update final rating**
>
> Dear Reviewer hFHk, please review the authors' response/updates, provide additional comments/questions if needed, and update your final rating. The discussion period ends tomorrow, March 14. Thank you!

---

### Official Review · Reviewer_v6qC · 2025-02-21

**Confidence:** 3
**Preliminary Rating:** 4
**Final Rating:** 5

**Summary:**

This paper introduces Federated Ordinal Learning which combines ordinal classification with federated learning. The authors evaluate two ordinal approaches (Binomial Cross-Entropy and Ordinal Encoding) against standard CE across four FL methods under increasing levels of client data heterogeneity. Results on the CSAW-M mammography dataset indicate that ordinal methods maintain better class ordering information when institutional data exhibits severe imbalance or missing classes.

**Strengths:**

- The paper introduces Federated Ordinal Learning (FOL), which appears to be the first systematic study combining ordinal classification with federated learning frameworks
- The authors use a controlled approach to simulate different levels of data heterogeneity, allowing for systematic evaluation of how ordinal methods perform under increasingly challenging conditions
- The use of multiple metrics provides a multi-faceted view of model performance
- I like the confusion matrices. They offer valuable insights into how different methods preserve ordinal relationships

**Weaknesses:**

- The authors fine-tuned hyperparameters only for the highest heterogeneity setting due to computational constraints
- Testing on only one dataset limits the generalisability of the findings
- While the paper provides empirical results, it lacks deeper theoretical analysis

**Detailed Comments:**

- The paper doesn't adequately address the computational costs of the different approaches
- A centralised (non-federated) learning baseline would provide context for how much performance is lost due to federation regardless of the ordinal approach
- Typo: "reseacht" instead of "research" on page 3
- Figure 4 (in the appendix) is referenced in the main text but without sufficient explanation of what insights can be derived from it

**Justification Of The Final Rating:**

The authors have added a runtime analysis which makes brings the work into real-world context and allows for more practical takeaways. They've also included a centralised baseline and provided details on the hyperparameter tuning process. Also, figure 4 is clearer now and brought better into context.

**Justification Of The Preliminary Rating:**

The methodology is sound with experiments that demonstrate how ordinal methods can maintain better class ordering in challenging federated scenarios with imbalanced or missing classes. However, limitations include hyperparameter optimisation for only the highest heterogeneity setting, evaluation on a single dataset and insufficient theoretical analysis. Overall though this is a good paper.

**Questions To Address In The Rebuttal:**

- How could hyperparameter tuning only for the highest heterogeneity setting have affected results at other heterogeneity levels?
- What evidence supports the generalizability of these findings beyond the single CSAW-M dataset used in the
- How do the computational costs compare between the different approaches, particularly for resource-constrained federated settings?
- Why wasn't a centralised learning baseline included to put the other results in better context? Would it be confusing?

---

> ### Author Response · Authors · 2025-03-07
>
> We thank the reviewer for their comments and suggestions. We have partially addressed some of their concerns in the main rebuttal comment. Below, we address the remaining ones.
>
> ### Computational costs
>
> We performed a runtime analysis to evaluate computational costs across methods. We report and comment on these results in Appendix  D.2 and Tables 3 and 4. We briefly touch upon these additional experiments in the Results section of the main text to refer the reader to the related Appendix.
>
> ### Centralised baseline
>
> To provide more context on the performance of ordinal approaches in centralised settings, We present the centralised baseline results for CE, BCE, and OE, trained using data splits from the lowest heterogeneity scenario (α = 10.00, p = 0.99), which best approximates a centralised setting. We report these results in Table 2 in Appendix D.1, which we refer to in the Results section of the main text.
>
> ### Additional comments on the results on only one dataset
>
> As this is the first evaluation of ordinal approaches in heterogeneous federated learning, we began our analysis using a dataset specifically designed as a benchmark for ordinal classification. The CSAW-M dataset not only was curated with ordinality in mind, but also offers additional advantages. All images originate from a single centre and are annotated using a consistent protocol, which minimizes potential confounding factors due to distribution shifts or variations in annotation strategies. Moreover, the labels in CSAW-M were deliberately designed to reflect ordinal relationships, providing a more robust basis for testing ordinal methods compared to naturally ordinal tasks, such as breast cancer classification using BI-RADS, where the labels are often noisy.
>
> Our choice of the CSAW-M dataset, combined with the Bernoulli-Dirichlet sampling strategy, allowed us to isolate the effects of imbalanced class distributions and missing classes. Additionally, the federated learning methods and ordinal strategies we investigated have been validated across diverse contexts in the literature, suggesting that their behaviour under heterogeneity is not specific to a single dataset. While we recognize the value of testing on multiple datasets, CSAW-M serves as a controlled and reliable benchmark for ordinal classification in medical imaging.
>
> Building on these results, we plan to extend our analysis to additional datasets and tasks. Due to the limited time available for the rebuttal, we are unable to provide these additional results at this stage. We have acknowledged this limitation in the Conclusions Section and have outlined the expansion of our analysis to other datasets as a direction for future work.
>
> We have also rephrased the comment to Figure 4 in the Results to highlight its contribution.

---

> > ### Comment · Reviewer_v6qC · 2025-03-12
> >
> > I thank the authors for addressing my concerns. The updated paper is stronger and I will be raising my score from 4 to 5.
> >
> > I would also like to commend the colour-coding in the update to make changes and the reasons/source for them much clearer. This was helpful.

---

### Author Rebuttal · Authors · 2025-03-07

**Rebuttal:**

We thank the reviewers for acknowledging the value and strenghts of our work and for their valuable comments and suggestions.

This rebuttal addresses the common concerns of reviewers v6qC and xAPJ. Individual concerns from each reviewer are addressed as seperate comments to each reviewer. We have modified and updated the manuscript to address the reviewers' concerns.
Modifications to the manuscript to implement reviewer v6qC's comments are highglighted in red, the ones for reviewer hFHk are highligheted in blue, and the ones for reviewer xAPJ are highlighted in green.

### Hyperparameter fine-tuning

Both reviewer v6qC and reviewer xAPJ asked for further clarification regarding the impact of having fine-tuned the hyperparameters only on the highest heterogeneity setting.

We tuned the hyperparameters in the most challenging (highest heterogeneity) setting to ensure robustness under severe non-IID conditions. This approach was chosen due to computational constraints and to guarantee that the model could handle the worst-case scenario. However, we acknowledge that the optimal values in the highest heterogeneity setting may not be ideal for lower heterogeneity scenarios. In lower heterogeneity settings, client data are more balanced and the degree of divergence between local models is reduced, which can change the sensitivity to these hyperparameters.  In Appendix C of the manuscript, we have provided a detailed explanation of how each of the considered FL methodologies can be impacted by this choice. Additionally, since we hypothesised that FedProx was the most impacted by this choice and reviewer xAPJ raised concerns about this, we have re-run the hyperparameter tuning for the middle heterogeneity setting for FedProx. Results of these additional experiments can be found in Table 1 in Appendix C. We have addressed these concerns and new results in the Results and Conclusions sections of the main text, and referred the reader to Appendix C.

### Results only on one dataset

As this is the first evaluation of ordinal methods in heterogeneous FL, we selected CSAW-M, a controlled benchmark specifically designed for ordinal classification. Its single-centre origin, consistent annotation protocol, and carefully structured ordinal labels minimize confounding factors, enabling a clear analysis of ordinality under client heterogeneity. However, we are planning to extend the analysis to other datasets and this is highlighted in the Conclusions section.

**Supporting Material:**

/attachment/bdfb6260270a2befef080be4960ce9a3e581b5e4.pdf

---

### Meta-Review · Area_Chair_irnW · 2025-03-20

**Recommendation:** Accept (Poster)
**Confidence:** 5

**Metareview:**

All reviewers agree that this work is a valuable contribution on integration of ordinal classification into federated learning for medical image analysis, noting the clear presentation, comprehensive systematic experiments evaluating different levels of data heterogeneity, and improved discussion of limitations/future directions post rebuttal.